# Research on hydrodynamic performance of S-type turbine based on linear wave

**LingJie Bao**[1], **Ying Wang**[1]*, **JunHua Chen**[2], **Hao Li**[1], **ChuHua Jiang**[2], **Yue Zhuo**[1], **Siru Wu**[1]

1 Faculty of Mechanical Engineering & Mechanics, Ningbo University, Ningbo, Zhengjiang, China, 2 College of Science and Technology Ningbo University, Ningbo, Zhengjiang, China

* wangyinghangzhou@163.com

## Abstract

The hydrodynamic performance of a Savonius type turbine (S-type turbine) in wave field is studied. The method of combining numerical simulation with physical experiment is adopted.Based on linear wave theory and turbulence model, Star CCM+numerical simulation software is used for digital modeling, and overlapping grid technology is used for grid modeling. Dynamic Fluid Body Interaction (DFBI) model is called to control the movement of S-type turbine, and second-order time discretization is adopted to establish a two-dimensional wave field model and conduct numerical simulation. At the same time, the physical test system of S-type turbine was developed for physical experiment verification, and the rotating performance of S-type turbine under different wave heights and pe-riods was measured, and the results of numerical simulation and physical experiment were com-prehensively evaluated. The results show that in wave field, the inverted S-shape arrangement of horizontal axis is better than the positive S-shape arrangement of horizontal axis and the vertical axis arrangement, and the captured energy is more than 3 times. At the same time, the fluctuating rotation speed of S-type turbine in wave field needs to be close to the wave frequency to capture more wave energy, which has guiding significance for practical engineering.

## 1. Introduction

The oceans are rich in renewable resources. Capturing tidal current energy and wave energy is of great significance to the exploitation of Marine resources. Domestic and foreign experts and scholars study the optimization of turbine structure to capture renewable energy.Narendra et al. [1] evaluated the performance of the improved turbine and compared it with the simple two-blaked Savonius turbine and some well-known Savonius turbine designs by using CFD software Fluent. Kumar et al. [2] discussed that the vertical axis turbine can start at a very low fluid speed in rivers and canals. Yao et al. [3] effectively improved the efficiency and stability of the water turbine by improving the basic parameters of the S -type water turbine and increasing the auxiliary agency. Kumar et al. [4] found that the efficiency of energy capture could be improved by changing the cross-section profile shape of the twisted Savonius rotor.

**Data Availability Statement:** All relevant data are within the manuscript and its Supporting Information files.

**Funding:** This research was co-funded by the following three projects: Ningbo 2035 science and

technology innovation Yongjiang key research and development project,2024Z266; Ningbo Natural Science Foundation Project, 2023J172; and Ningbo Municipal Major Science and Technology Research and Unveiling Project, 20212ZDYF020015.

**Competing interests:** The authors declared no potential conflicts of interest with respect to the research, authorship, and/or publication of this article.

Ostos et al. [5] introduced a novel configuration to improve the performance of traditional Savonius blades. Salleh et al. [6] improved power performance by changing the longitudinal position of the pushing blades and returning blades.Mao et al. [7] studied the influence of blade arc Angle on the performance of a typical two-blade Savonius wind turbine, which improved the power generation efficiency of the turbine by using transient computational fluid dynamics method.Kerikous et al. [8] performed a number of transient computational fluid dynamics (CFD) simulations using Star-CCM+ to maximize the output power of the Savonis turbine by modifying 12 parameters of the blade profile. Dos Santos et al. [9] developed a computational model to study the savonius turbine turbulent flow problem,and simulate an oscillating water column device. Li et al. [10] studied vertical axis wind turbines. The force distribution of the blades under different vertical axis turbine configurations is obtained, and the torque coefficient and power coefficient are obtained. The wake impact of nearby objects on the vertical axis wind turbine is analyzed. Tutar et al. [11] studied the generation and propagation of conventional water waves and the performance of Savonius rotor with three blades on its horizontal axis.Talukdar et al. [12] found that resistance Savonius fluid power turbine (SHT) has the huge potential of small-scale power generation with free flowing water. Patel et al. [13–16] studied the influence of overlap ratio, aspect ratio, flow velocity and geometrical parameters of the water channel on the hydrodynamic performance of the turbine. A theoretical study for predicting the hydrodynamic performance of vertical Savonius turbines is also introduced.Wu et al. [17] proposed a one -way opening valve -type Savonius rotor design. The results showed that the curvature radius of the blade was large and the opening rate was small.

Yao et al. [18] studied the maximum power coefficient and tip speed ratio of S-type turbines under tidal flow energy by changing different geometric parameters. Rizki et al. [19] studied that Savonius turbine has a high power coefficient (CP) in the range of low tip speed ratio (TSR), and CP decreases with the increase of TSR. Wu et al. [20] studied the rotation mechanism of the S -type water turbine in the wave field in the form of vertical shaft layout. Huang et al. [21] studied the application of the S-type turbine in the form of a horizontal axis layout on the floating breakwater, and completed the practice of energy capture of S-type turbine in wave field. Li et al. [22] studied wave energy capture of S-type hydraulic turbines under different control strategies, mainly from different installed phase angles, different wave cycles, and different control speeds.

Based on the research of domestic and foreign scholars on S-type turbine, the performance of S-type turbine under air flow condition is studied mainly by improving its shape parameters. In recent years, more and more scholars have studied and analyzed the hydrodynamic performance of S-type turbines in tidal energy capture.And S-type turbines are mainly used in wind field and low velocity flow field. The motion path of water particles in the flow field is smooth. However, the water particles of the wave may be elliptical or circular depending on the depth and wave characteristics. As shown in Table 1, S-type turbines have different characteristics of capturing energy in different scenarios.The discussion about whether S-type turbine is suitable for wave field, and the research on the rotational performance and energy capture

**Table 1. Comparative analysis of key points of S-type turbine.**

| Model | Application scenarios | Characteristic | Energy |
|-------|----------------------|----------------|--------|
| S-type turbine | Wave field | The velocity and direction of water particles change in a wave field | Wave energy |
| | Flow field | The velocity and direction of water particles in a pure flow field are constant | Tidal current energy |
| | Wind field | | Wind energy |

performance of S-type turbine in wave field are relatively few.Therefore, this paper studies the energy capture performance based on S-type turbine. By studying the layout of S-type turbine, changing the depth of wave field of S-type turbine into water, and studying its hydrodynamic performance under different wave conditions. The results have a certain reference value for guiding the application practice of S-type turbine.

## 2.Geometry and property parameter definitions

### 2.1 geometric model

The S-type turbine has two semi-circular blades and two end plates. The three-dimensional model and physical drawings of the S-type turbine are shown in Figs 1 and 2. It has a simple structure and is widely used in flow field, including wind field and flow field, but relatively few applications in wave field. The S-type turbine is a resistance type runner driven by the pressure difference between the front and rear blades. In deep water wave condition, the motion path of water particles is circular. Water particles drive the S-shaped blades, which turn the turbine, as shown in Fig 3. According to the wave data and engineering application background in the middle and far seas, combined with the size of the experimental tank and the generated wave-form, the geometric ratio of the experimental model is 1:5. The geometric characteristics and hydrodynamic parameters of the S-type turbine are shown in Tables 2 and 3.

### 2.2 S-type turbine energy conversion model

When a S-turbine captures tidal energy, its hydrodynamic performance is usually judged by power coefficient, torque coefficient and tip speed ratio because the water flow is uniform. However, for the study of S-turbine under the action of waves, the speed and direction of

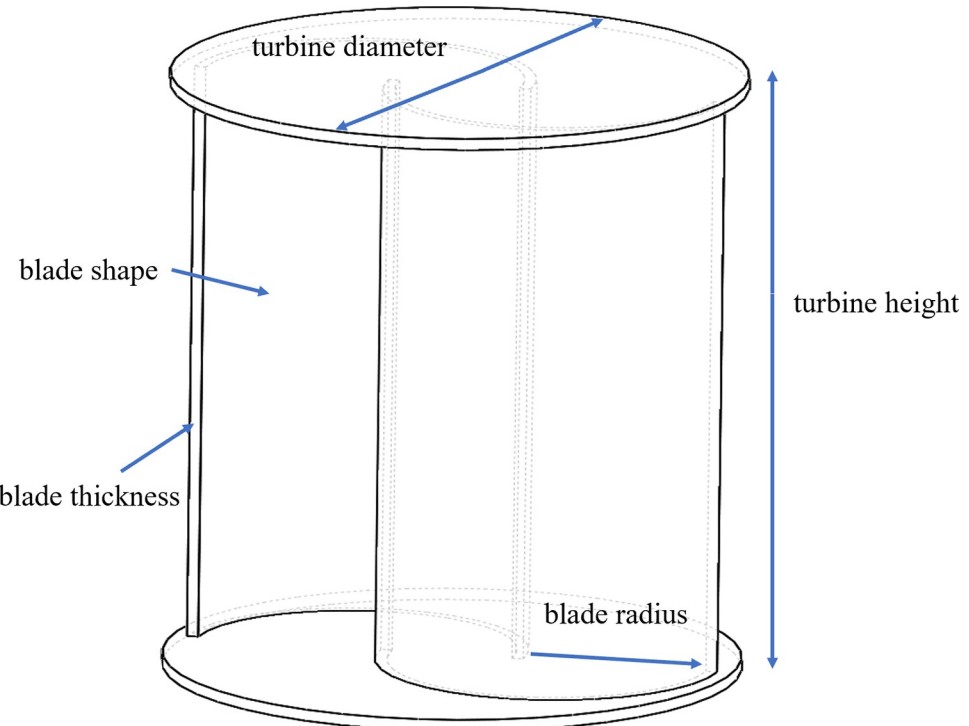

**Fig 1. Three -dimensional model.**

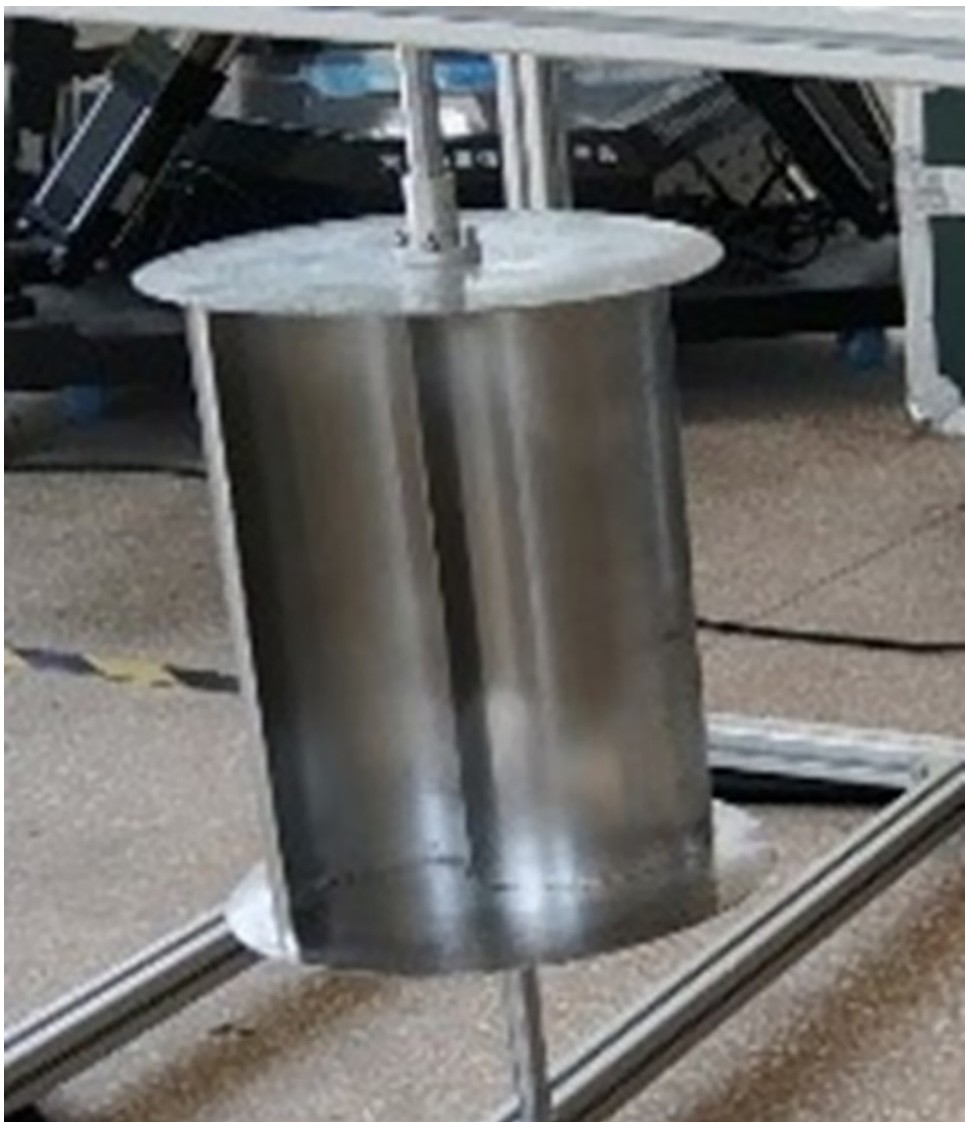

**Fig 2. Physical model of the blade group.**

water particles change all the time. Therefore, energy is used to judge the performance of S-turbine.From the principle of conservation of energy, the generated wave energy (incident wave energy) is converted into reflected wave energy, transmitted wave energy, loss of wave energy and kinetic energy to drive S-type turbine. As shown in Eq (1). [23].

$$P_{\text{wave}} = P_c + P_r + P_t + P_s \tag{1}$$

Where $P_{\text{wave}}$ is the incident wave power, $P_c$ is the dissipation power, $P_r$ is the reflected wave power, $P_t$ is the transmitted wave power, $P_s$ is the S-type turbine power [23].

Incident wave power formula

$$P_{\text{wave}} = \frac{1}{8}\rho g H_i^2 C_g = \frac{1}{32\pi}\rho g^2 H_i^2 T \tag{2}$$

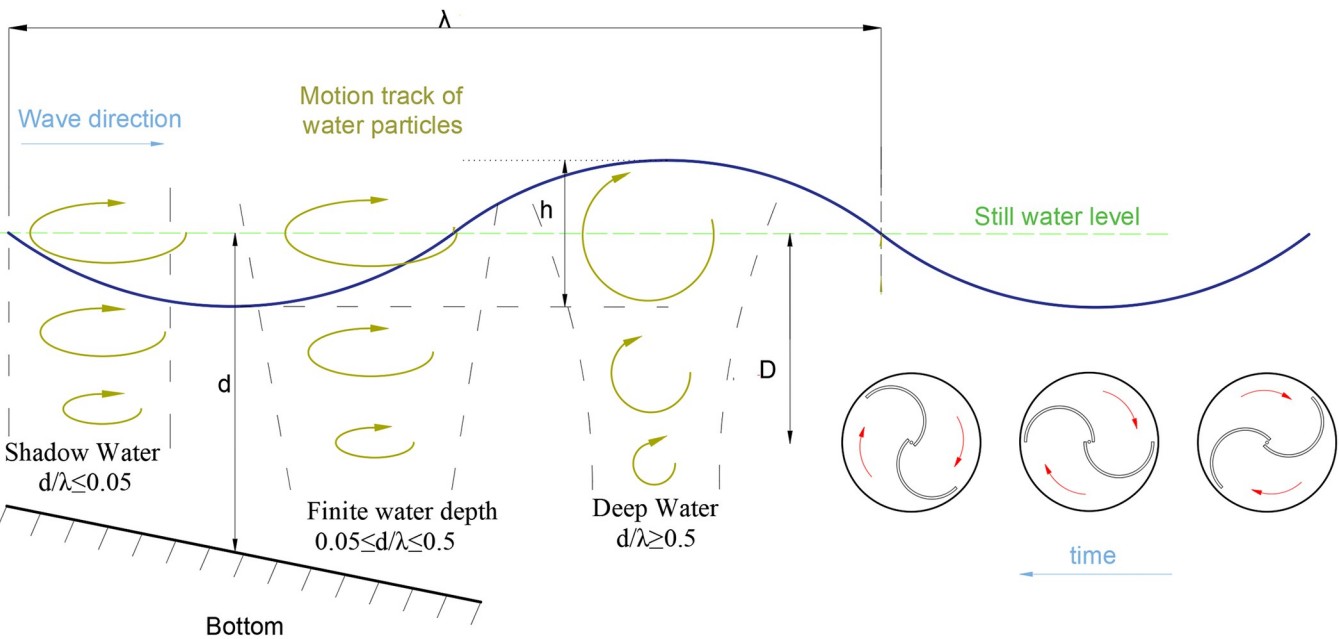

**Fig 3. Water turbine movement diagram.**

Where $\rho$ is the water density, $g$ is the gravitational acceleration, $H_i$ is the incident wave height, $C_g$ is the group velocity, $C_g = \frac{1}{2}c$, c is wave velocity, $c = \frac{gT}{2\pi}$, $T$ is the incident wave period.

Reflected wave power and transmitted wave power are related to wave height. To calculate the power of S-type turbine, its moment of inertia and its angular velocity need to be measured. Therefore, it is necessary to install sensors in physical experiments and set their motion parameters in simulation experiments

The formula for calculating the moment of inertia of the S-type turbine is as follows

$$
\begin{aligned}
I = J_{\mathrm{t}} &= J_{\mathrm{i}} + J_{\mathrm{d}} \\
&= 2 \times \left( \frac{1}{2} \int_0^{2\pi R} \lambda R^2 dl + m_i \bullet l^2 \right) + 3 \int_0^R 2\pi r^3 h\rho dr \\
&= m_i \bullet \left( \frac{d}{2} \right)^2 + 2m_i \bullet \left( \frac{d-e}{2} \right)^2 + 3m_d \bullet \left( \frac{D_d}{2} \right)^2
\end{aligned}
\tag{3}
$$

**Table 2. S -type water turbine geometric characteristics.**

| Parameter | Value | Unit |
|---|---|---|
| Turbine diameter | 0.5 | m |
| Blade radius | 0.15 | m |
| Blade thickness | 0.002 | m |
| Turbine height | 0.61 | m |
| Blade shape | Semicircle | / |
| Overlap rate | 0.15 | / |
| Turbine volume | 0.0006 | $m^3$ |
| Moment of inertia of turbine | (0.05, 0.09,0.11) | $kg \cdot m^2$ |

**Table 3. Experimental condition parameter table.**

| Parameter | Symbol | Value | Unit |
|---|---|---|---|
| Depth of water | $D_1$ | 2 | m |
| Sink length | L | 70 | m |
| Sink width | W | 3.8 | m |
| Sink depth | $D_2$ | 3 | m |
| Wave height | h | 0.03,0.06,0.09,0.12 | m |
| Wave period | T | 1.4–2.0 | s |

Where $m_i$ is the turbine mass, $m_d$ is the end disc mass, $d$ is the blade diameter, $e$ is the overlap ratio, $D_d$ is the end disc diameter.

The formula for calculating the rotational energy of the S-type turbine is:

$$W_s = \frac{1}{2} I \omega^2 \tag{4}$$

Where, $W_s$ is the energy of the turbine, $I$ is the moment of inertia of the turbine, and $\omega$ is the speed of the turbine.

The formula for calculating the power of the S-type turbine is:

$$P_s = \frac{\mathrm{T}n}{9550} \tag{5}$$

The capture efficiency of the turbine is $C_P$

$$C_P = \frac{P_s}{P_{wave}} = \frac{W_s}{W_{wave}} \tag{6}$$

The remainder is dissipated power

$$P_c = P_{wave} - P_r - P_t - P_s$$
$$= \frac{1}{32\pi} \rho \mathrm{g}^2 (H_i{}^2 - H_r{}^2 - H_t{}^2) - \frac{\mathrm{T}n}{9550} \tag{7}$$

Where $T$ is a torque and $n$ is a rotating speed.

## 3. Numerical simulation

### 3.1 Simulation domain size and boundary conditions

In order to prevent the flume boundary from interfering with the result, the data distance needs to be set sufficiently large. The length of the numerical simulation area is 30 λ and the width is 2 L. According to the experimental specification, the waveform is relatively flat at the distance of 10 λ. Therefore, the S-type turbine is set 10 λ away from the left boundary, as shown in Fig 4. At the same time, typical boundary conditions are adopted. The speed inlet is set at the left boundary. The right boundary is set as the pressure outlet, and the damping wave elimination is set, and the wave elimination length is 2 λ, to prevent the interference of the reflected wave on the S-type turbine. The upper and lower boundaries are set as symmetric planes, and the S-type turbine is centered to avoid boundary conditions affecting the experimental results. The S-type turbine is set to the wall. The incident wave is linear.

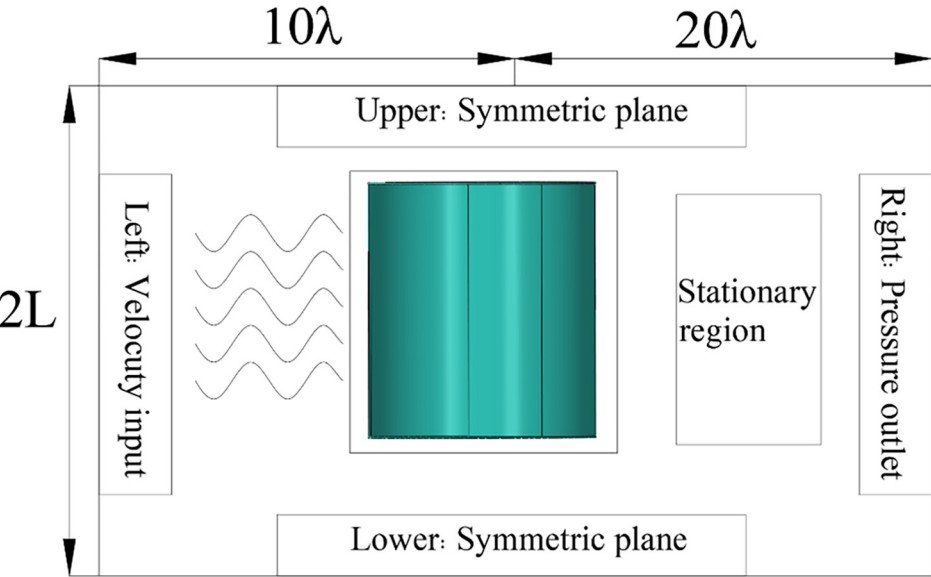

**Fig 4. Computational domain and boundary conditions.**

## 3.2 Meshing and simulation model selection

In the solution calculation, because the S-type turbine is rotating, the calculation domain grid is divided into two parts (background grid and overlapping grid). As shown in Fig 5A. Tangential mesh and prismatic layer mesh are used for background mesh, and polyhedral mesh and prismatic layer mesh are used for overlapping mesh. The polyhedral grid of overlapping

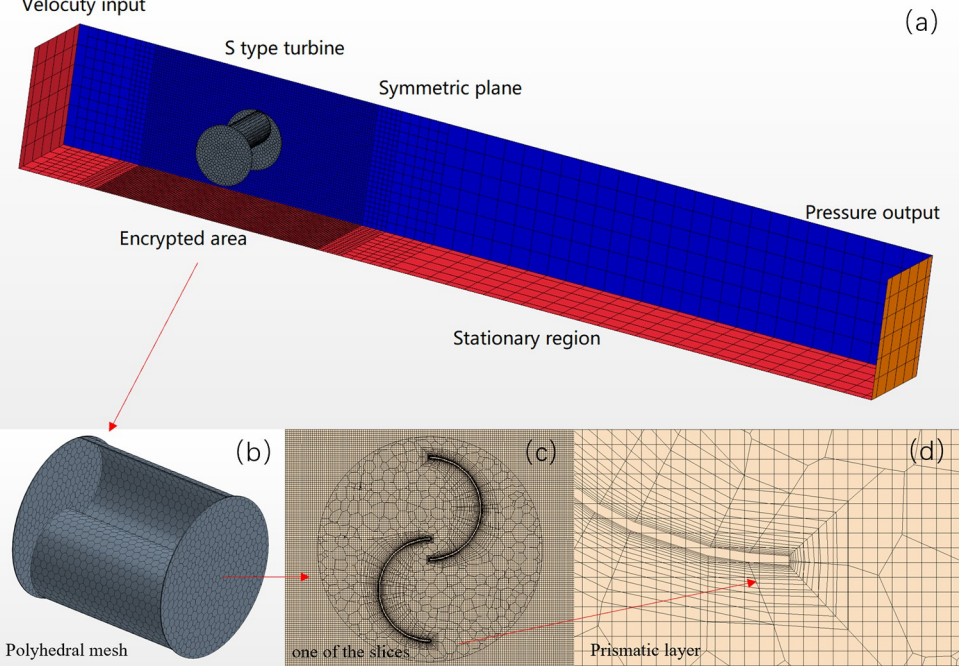

**Fig 5. Fluid domain meshing diagram.**

**Table 4. Details of mesh refinement levels.**

| Refinement level | No. of cells | No. of faces | No. of Verts | Max $C_p$ | Max TSR |
|:---:|:---:|:---:|:---:|:---:|:---:|
| 1 | 21081 | 59843 | 23541 | 0.09 | 0.86 |
| 2 | 28824 | 81832 | 32150 | 0.14 | 0.9 |
| 3 | 46570 | 132186 | 50875 | 0.18 | 1 |
| 4 | 70201 | 202470 | 77035 | 0.19 | 1 |

regions is shown in Fig 5B. One of the slices is shown in Fig 5C. In order to ensure the precision of interface interpolation and reduce the numerical transmission error, 16 layers of prisms are arranged at the blade, as shown in Fig 5D. In order to verify the independence of the grid, the size of the first layer of grid is adjusted, and the thickness of the first layer is set to 1mm, 0.8mm, 0.4mm, 0.2mm respectively. The difference between the calculated results of the average power coefficient (TSR = 0.9) of the S-type turbine under the best working conditions is less than 2%. Finally, the thickness of the first layer of mesh was set to about 0.2mm, the growth rate was set to 1.2, and the y+ value of the blade wall was controlled between 60 and 144. The number of overlapping grids is 71783, the number of background grids is 5955364, and the total number of calculation domain grids is set to 6027147. The residual of each numerical calculation is lower than $10^{-5}$, which is in the convergence of the calculation, otherwise the software will make errors, which is easy for the operator to adjust.

Grid division starts with the coordinate (0,0,0) as the center, considering the size of the first layer of basic grid will affect the overall grid number, but also to improve computing efficiency. At the same time of mesh refinement, the maximum power coefficient and maximum tip speed ratio were studied, as shown in Table 4. After the third adjustment, $C_p$ almost reached a constant level, and TSR was basically stable. Thus, the grid independence is achieved. This level of refinement corresponds to 70,201 units, 202,470 faces, and 32,150 vertices.

Reynolds Average Navid Stokes (RANS) standard k-ε model was used to simulate turbulence, and high-wall processing function method was used. Choose an implicit unsteady model. Euler multiphase flow has two phases, including air and water. The multi-model global protection method across overlapping grid interfaces is improved by using overlapping multinomial protection. In the spatial discretization, the momentum and turbulence of the governing equation adopt the second-order upwind scheme. Time dispersion is a first order implicit scheme. Each time step is 0.01s, and each time step is iterated 15 times and calculated 30s. Ensure that the rotation Angle of the turbine is less than 1˚ in a time step. The semi-implicit pressure correlation equation (SIMPLE) algorithm is used to solve the pressure-velocity coupling numerically. Each time step iterates 15 times. The standard for residual convergence is $10^{-5}$.

## 3.3 Simulation validation

Based on the simulation research of S-type turbine, the physical experiment was carried out in Ningbo Institute of Technology of Zhejiang University.The physical experiment of S-type turbine is carried out in a tank of 70 m in length, 4 m in width and 3 m in depth. The left side of the tank is provided with a wave making system. A wave elimination net is set on the right side of the tank to absorb transmitted waves and reduce the generation of reflected waves. Dynamic torque sensor, whose model number is DYN201, is driven by synchronous wheel and synchronous belt to collect speed and torque of S-type turbine. The S-turbine test device is located at more than ten wavelengths of the wave generator. The resulting wave can be observed to be stable. The experimental framework is shown in Fig 6. The physical experimental tank is shown in Fig 7. The absorbing beach is shown in Fig 8.

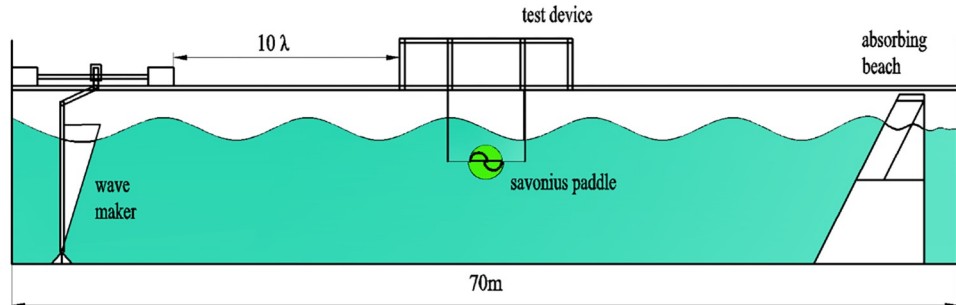

**Fig 6. Experiment framework.**

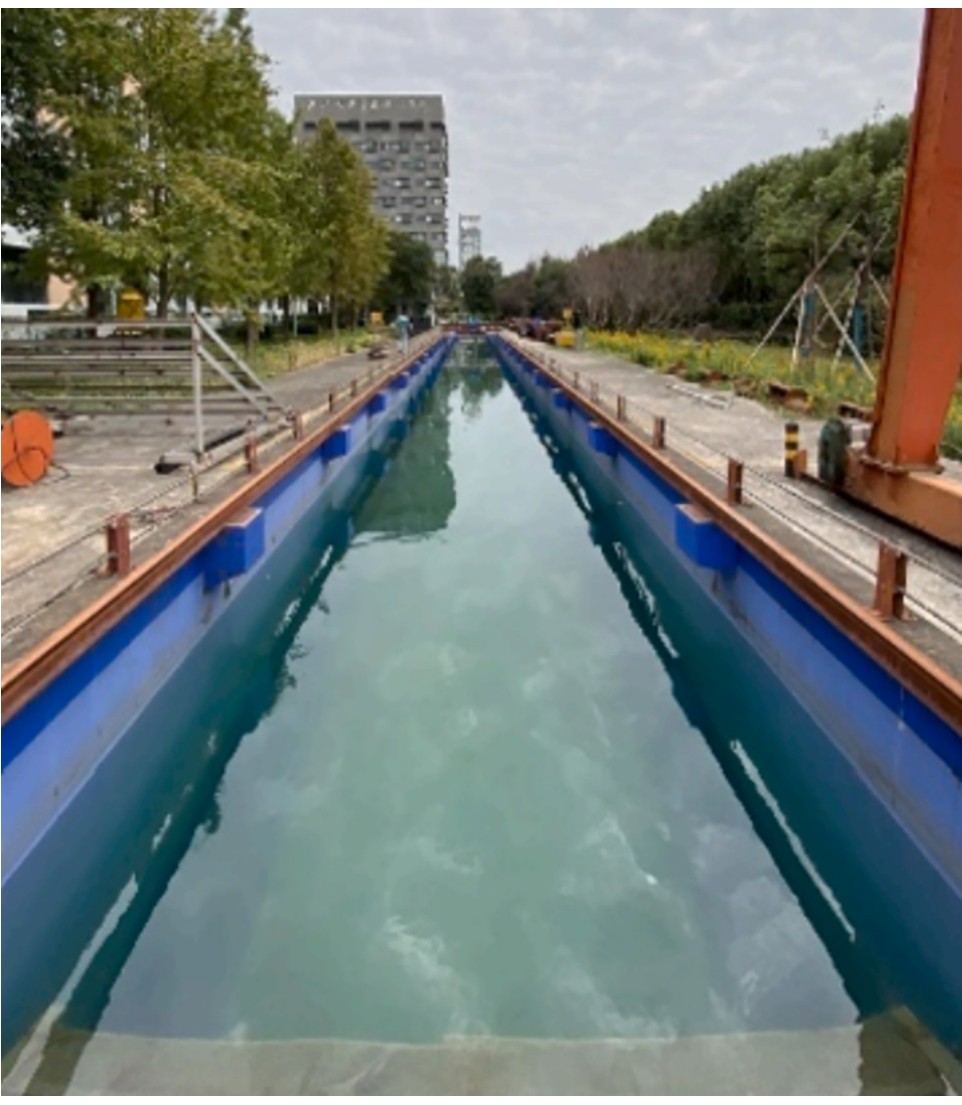

**Fig 7. Physical experiment tank.**

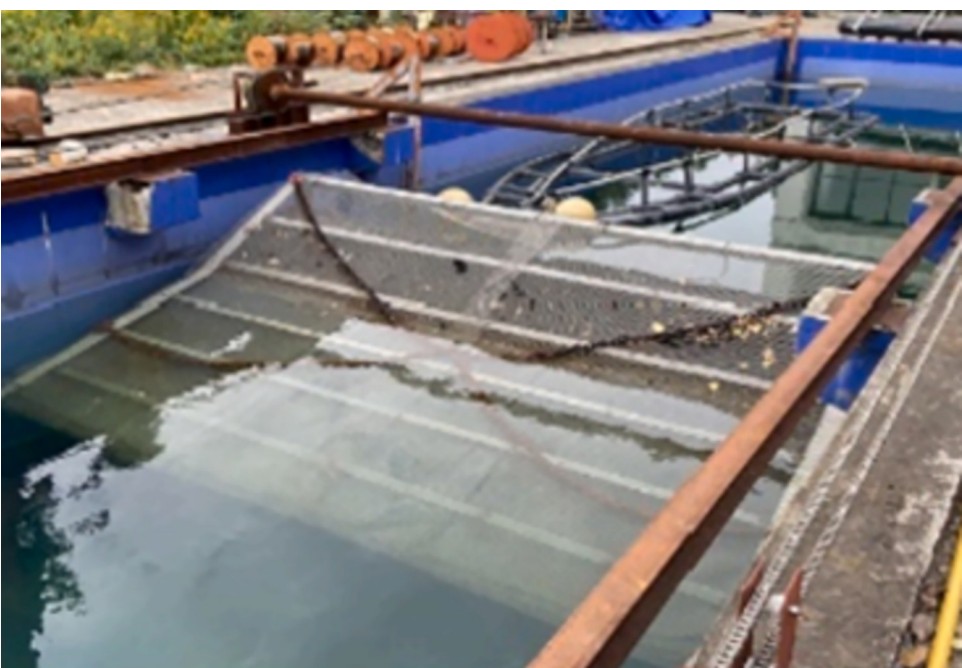

**Fig 8. Absorbing beach.**

Pool wave is linear wave with different period. S-type turbine submerged in water. By changing turbine layout, the performance of S-type turbine under different working conditions is analyzed. Wave-making is done with a shake plate, as shown in Fig 9. Servo motor drives ball screw to move back and forth. The shaker pushes the wave to produce a stable waveform. Due to the boundary effect and the viscosity and gravity of water, the wave-making of shaken plate will have waveform attenuation and deformation. Therefore, data collection begins after 10 stable waveforms. The experimental arrangement is shown in Fig 10.

## 4 Results analysis

The main evaluation indicators of the hydrodynamic performance of the S-type turbine are the angular velocity of the S-type turbine in the wave field and the kinetic energy obtained by the S-type turbine. The rotational angular velocity and the obtained energy of the S-type turbine in the wave field are one of the important indicators to measure the energy conversion of the S-type turbine. The greater the rotational speed obtained by the S-type turbine in the wave field, the better the adaptability of the S-type turbine to the flow velocity of the wave field, which is more conducive to the capture of wave energy. The greater the energy obtained by the S-turbine, the higher the efficiency of wave energy conversion. This study mainly focuses on the effects of S-turbine layout, relative water entry depth D/h, different wave periods in wave field and external factors on the hydrodynamic performance of S-turbine.

### 4.1 Influence of arrangement of S-type turbines

For the research on the arrangement of S-type turbines, the vertical axis installation method and the horizontal axis installation method of the S-type turbine are studied, followed by the positive S-type installation method and the reverse S-type installation method facing the wave surface.

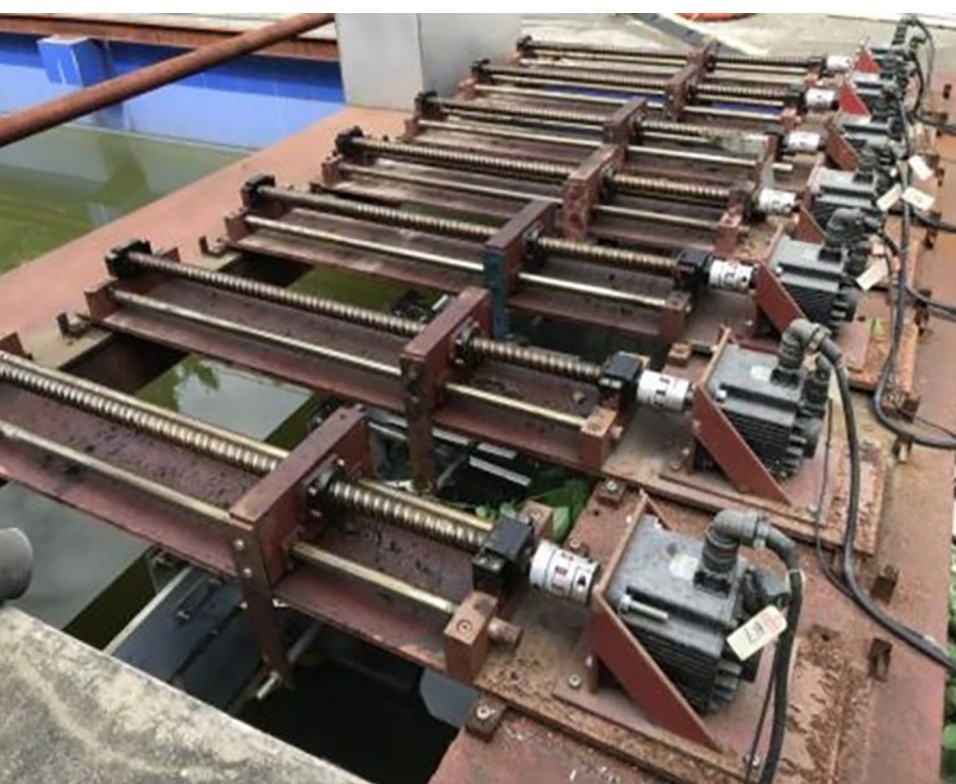

**Fig 9. Wave-making system.**

The vertical axis installation and the horizontal axis installation of the S-type turbine determine the difference in the force surface of the S-type turbine. The horizontally arranged and vertically arranged S-type turbines were studied in the wave field with wave heights of 0.06 m, 0.09 m and 0.12 m, as shown in Figs 11 and 12.

The rotational angular velocity function is set on the simulation software to obtain the rotational speed of S-type turbine. By writing a custom function, the energy of the S-type turbine is calculated, mainly the kinetic energy of the turbine. The calculation formula is shown in Eq (4). The angular velocity is measured by dynamic torque sensor in physical experiment. Through the comparison of the energy captured by the horizontally arranged S-type turbine and the vertically arranged S-type turbine, the experimental results are shown in Fig 13. Under the same c onditions, the horizontal S-type runner can capture the energy of the wave well. A horizontal S-turbine captures more than twice as much energy as a vertical turbine. At the same time, the higher the wave height, the greater the energy captured by the S-turbine. The energy capture at 0.12m is 30% more than that at 0.09m. After the S-type turbine is stabilized, the captured energy changes periodically.

To study the opening orientation of the wave facing side of S-type turbines, as shown in Figs 14 and 15, the S-type turbines were installed forward and reversely.

The energy capture under different wave fields is compared and analyzed. As shown in Fig 16, the higher the wave height, the greater the energy capture of the reverse S-type arrangement and the positive S-type arrangement. Moreover, the energy capture of the reverse S-type arrangement is significantly greater than that of the positive S-type arrangement. Because the reverse S-type arrangement accords with the movement path of water particles in the wave field, it is easier to drive the S-type turbine to rotate. Therefore, the energy obtained by the

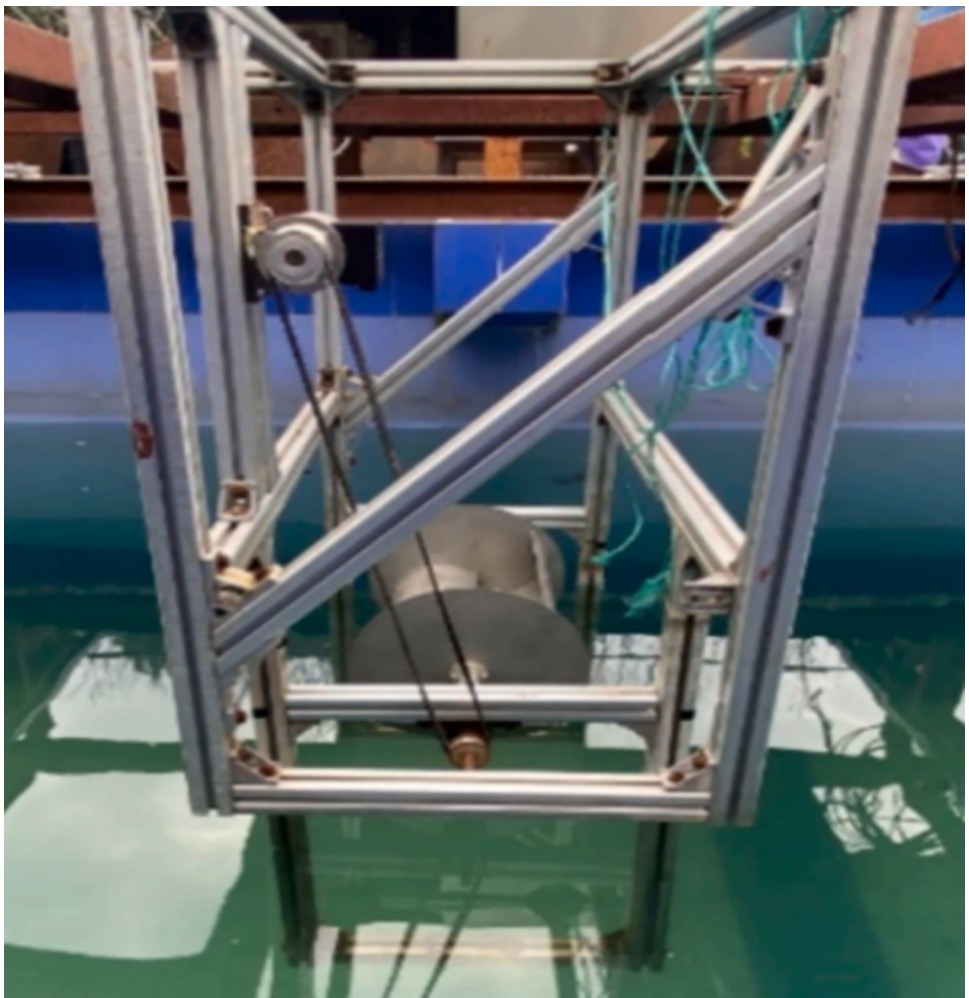

**Fig 10. Testing site.**

Reverse S-type turbine is more than twice that obtained by the positive S-type turbine in the experiment.

For the research on the rotation speed of the forward and reverse arrangement of the S-type turbine, as shown in Fig 17, the reverse S-type turbine on the wave-facing side has a faster rotation speed than the forward S-type turbine, and the higher the wave height, the more obvious the maximum value of the rotation speed, the greater the speed change, the greater the speed fluctuation. In the scheme of forward and reverse S-shaped arrangement on the seaward side, under the conditions of wave heights of 0.06 and 0.03, it has better periodicity, but the reverse rotation speed is greater than the forward rotation speed. The energy captured at the wave height of 0.06 m is obviously higher than that at the wave height of 0.03 m. The test condition with wave height of 0.06 m was verified by physical experiment. After the wave is stabilized and the S-type turbine is running smoothly, the experimental data of physical experiment and simulation experiment are shown in Fig 18. In the physical experiment, due to the viscosity of water, the error of turbine installation, the friction of bearing and transmission efficiency, the energy measured in the physical experiment is slightly smaller than that in the simulation experiment.

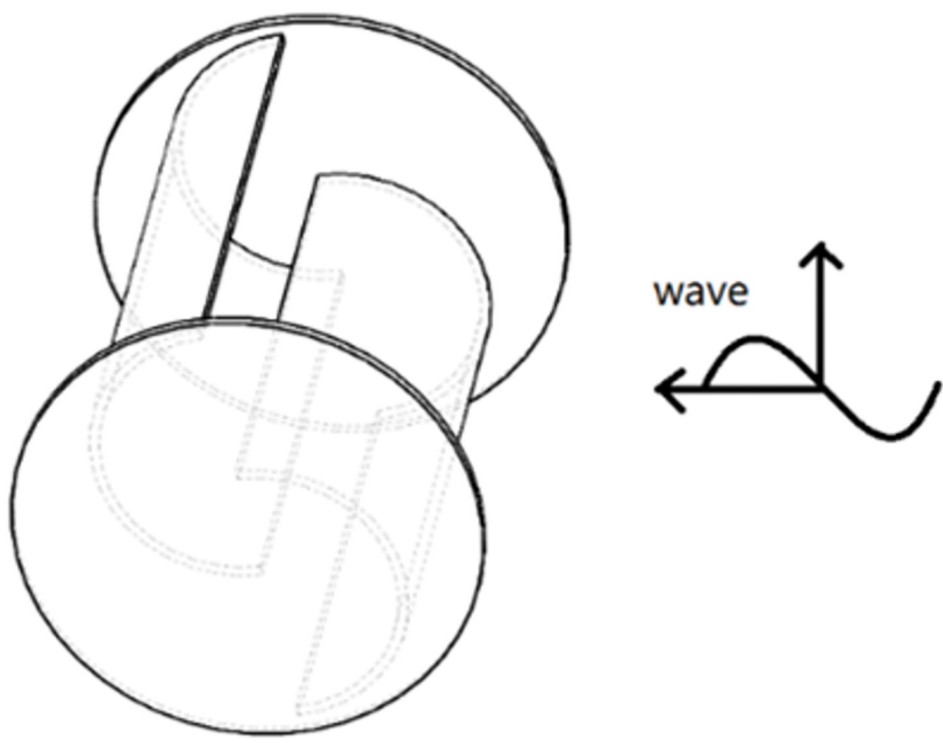

**Fig 11. Horizontal arrangement of S-type turbines.**

## 4.2 Influence of relative water entry depth D/h

For the study of the S-type turbine in the wave field, the relative water entry depth is a dimensionless research factor, and the relative water entry depth is the ratio of the water entry depth to the wave height. According to the research of literature 23 [23], this paper selects the water entry depth of about 0.75d for research. When the water entry depth is unchanged and D = 0.2 m, the wave height h is changed, as shown in Table 5. Observe the rotation of the S-shaped runner at different water depth relative to wave height.

The experimental results are shown in Fig 19. Under the condition of a certain depth of water, the speed of S-type turbine changes greatly with the change of wave height. When h = 0.12 m and 0.09 m, the rotational speed is relatively large, but the velocity changes are unstable. When h = 0 m, the S-turbine is almost impossible to spin. Under the condition of h = 0.6m, the rotational speed is relatively stable and relatively large.

At the same time, to verify the optimal water entry depth, the rotation of the S-type turbine in the wave field with the wave height of 0.06 m was studied when the water entry depth was 0.15 m, 0.2 m, 0.25 m, and 0.3 m, as shown in Fig 20.

Under the condition of a certain wave height, the water entry depth of S-type turbine is changed, and the relative wave height rotation of S-type runner is observed under different water entry depth. Where the water depth is 0.15m and 0.2m, the S-type turbine speed is larger. When the water depth is 0.25m and 0.3m, the speed of S-type turbine is small. It can be seen from the standard deviation of speed that the speed at 0.2m water depth is more stable than that at 0.15m water depth. When the wave is high, due to the viscosity of water, the deeper

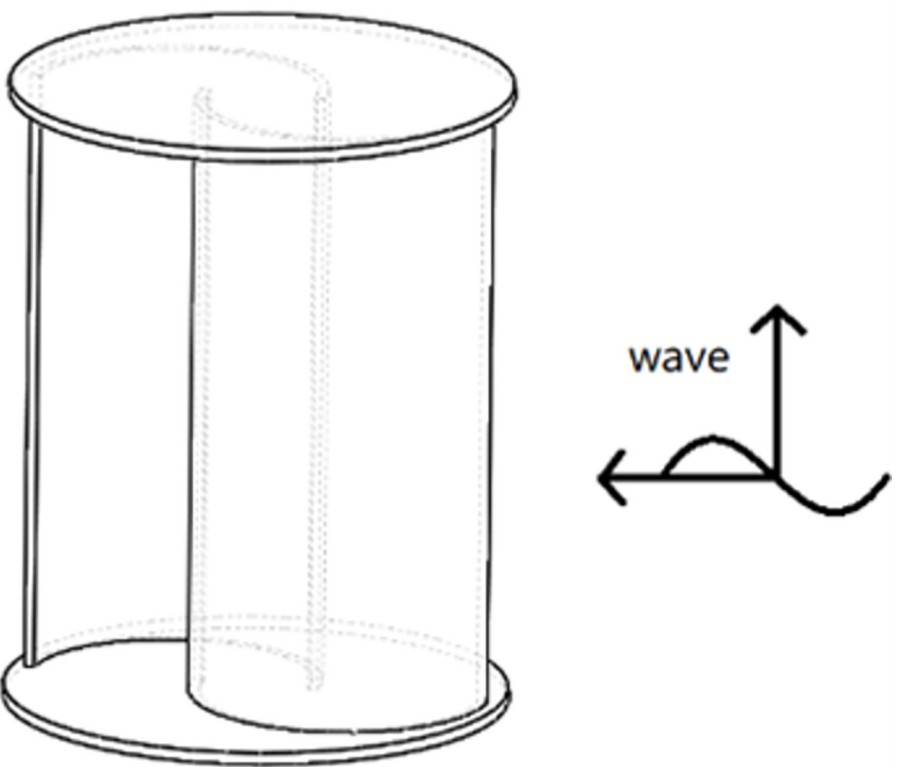

**Fig 12. Vertical arrangement of S-type turbines.**

the installation depth of S-type turbine, the speed is significantly reduced. When the installation depth of S-type turbine is shallow, the speed fluctuation is large.

## 4.3 The effect of period

Since the S-type turbine is composed of two blades, in the process of its rotation, the wave crest will inevitably impinge the wave energy on the blades of the S-type turbine, driving the S-type turbine to rotate. Due to the particularity of its structure and the motion of water particles in the theory of waves, the rotation movement of the S-type water turbine is related to the wave cycle.To study the impact of the wave cycle on the S -type water turbine, the first -class S -type water turbine will be used to study through physical experiments.

The first-stage S-type turbines were experimentally studied in the wave field with wave periods of 1.4 s, 1.6 s, 1.8 s, and 2 s, respectively. The experimental results are shown in Fig 21. As can be seen from the figure, the wave frequency decreases as the wave period increases in turn. The wave frequency is about twice that of an S-turbine. At the same time, the frequency difference is minimum when the wave period is 1.8 s. It is more accurately verified that each wave drives one blade on the S-turbine, and the most suitable wave period is 1.8 s.

## 5.Discussion

Based on the study of the key factors in wave field of S-type turbine, the energy capture experiment is carried out under different wave conditions with the wave facing side as the anti-S

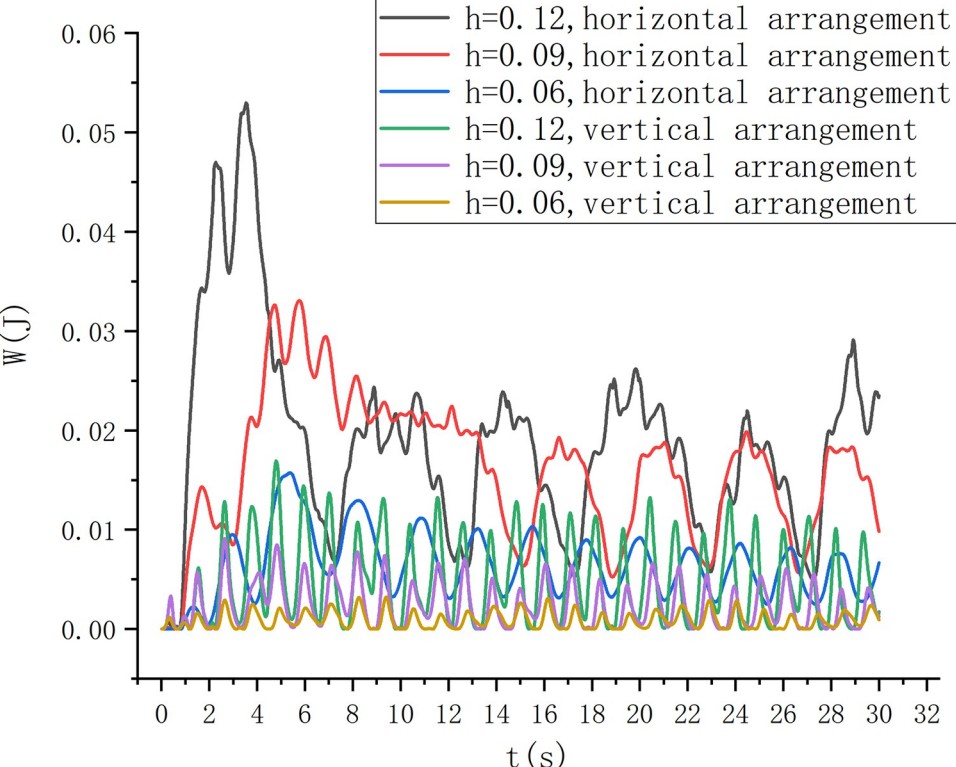

**Fig 13. Energy diagram of simulation experiment.**

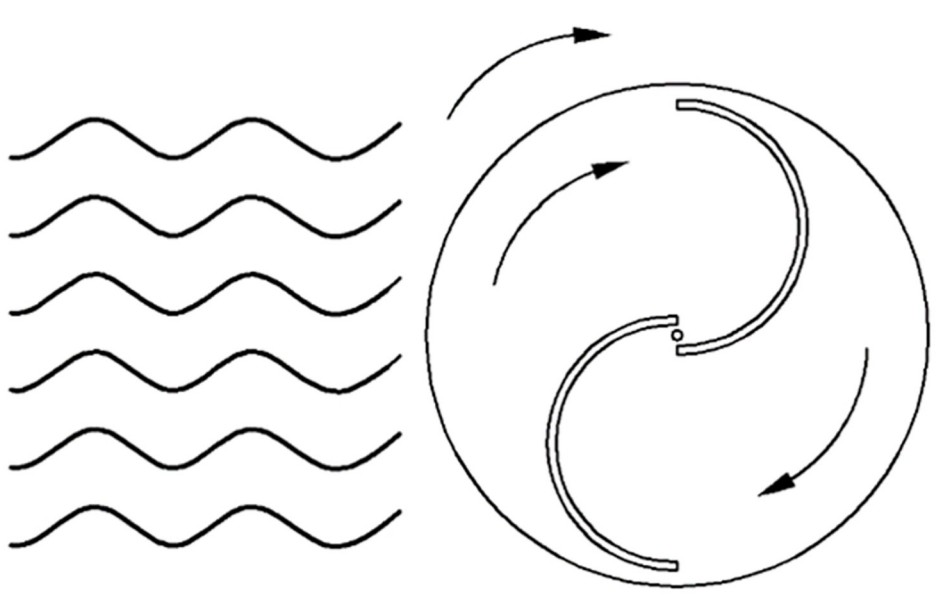

**Fig 14. Reverse S-type water turbine.**

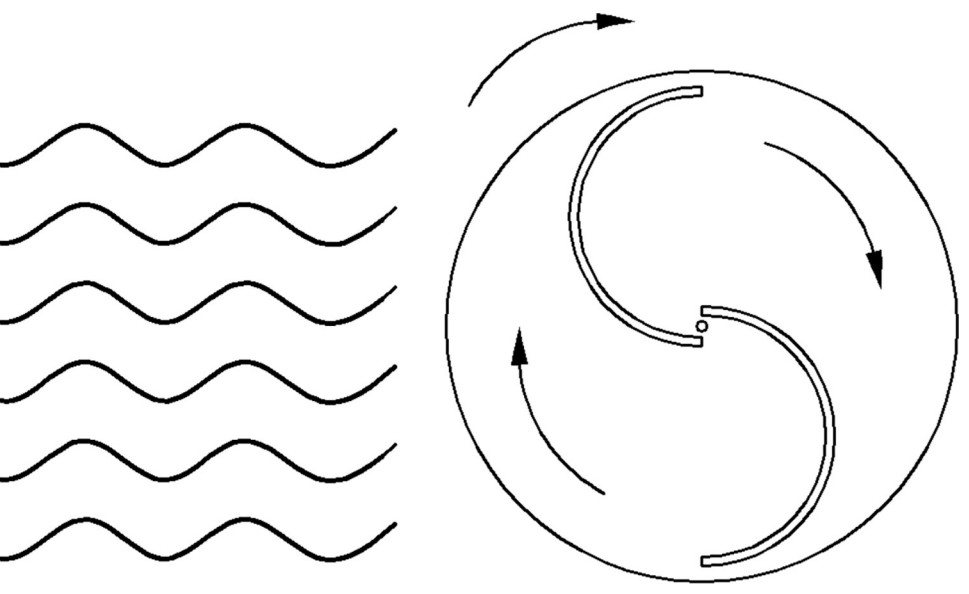

**Fig 15. Positive S-type water turbine.**

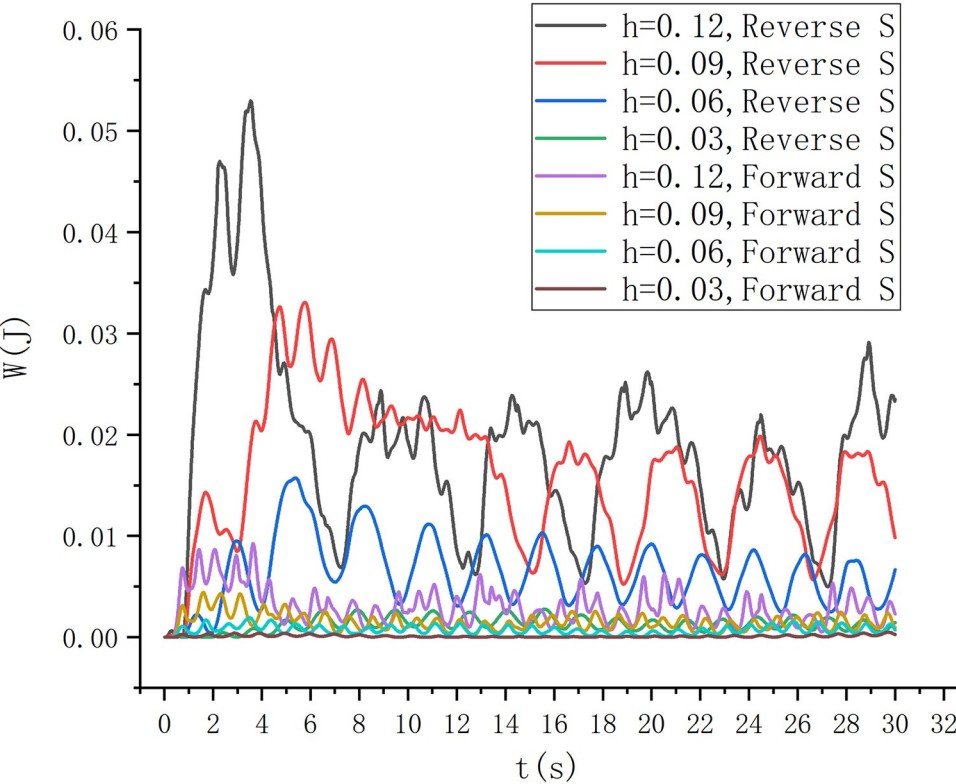

**Fig 16. Energy diagram of simulation experiment.**

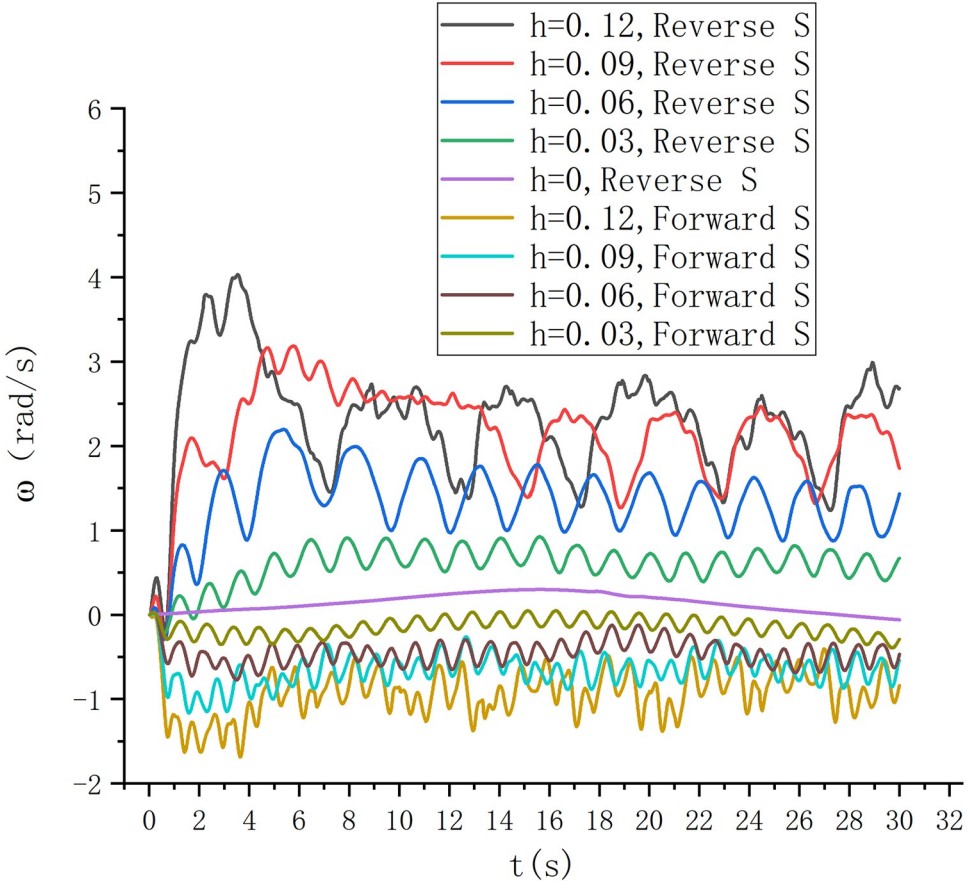

**Fig 17. Rotation diagram of physical experiment.**

horizontal axis. The daily energy capture diagram of a single S-turbine under the action of wave field is shown in Fig 22.

The calculated results are presented in the form of daily energy generation. It can be seen from the data that compared with the other two arrangements, the energy capture effect of the horizontal axis reverse S-type arrangement is significant. The captured energy decreases with the decrease of wave height.

At the same time, in order to compare the hydrodynamic performance of the S-type turbine, its characteristics are evaluated using the maximum $C_p$ and the maximum TSR, as shown in Table 6.

The maximum TSR of the horizontal axis and vertical axis arrangement of S-type turbine is different under the action of wave, the main reason is whether the operation mode of S-type turbine conforms to the motion trajectory of wave particles. But the energy capture coefficient is an intuitive index to measure whether wave energy is captured. The energy capture effect of the S-type turbine with horizontal axis arrangement is better than that of the positive S-type turbine in wave field. In the wave field, the effective impact duration of the wave is short, and the maximum TSR is relatively small, which is the main reason for the low energy capture efficiency of the S-turbine with vertical axis arrangement.

## 6. Conclusions

The operation of modern Marine facilities is increasingly dependent on energy. The offshore turbine can effectively capture and transform wave energy and promote the development of S-

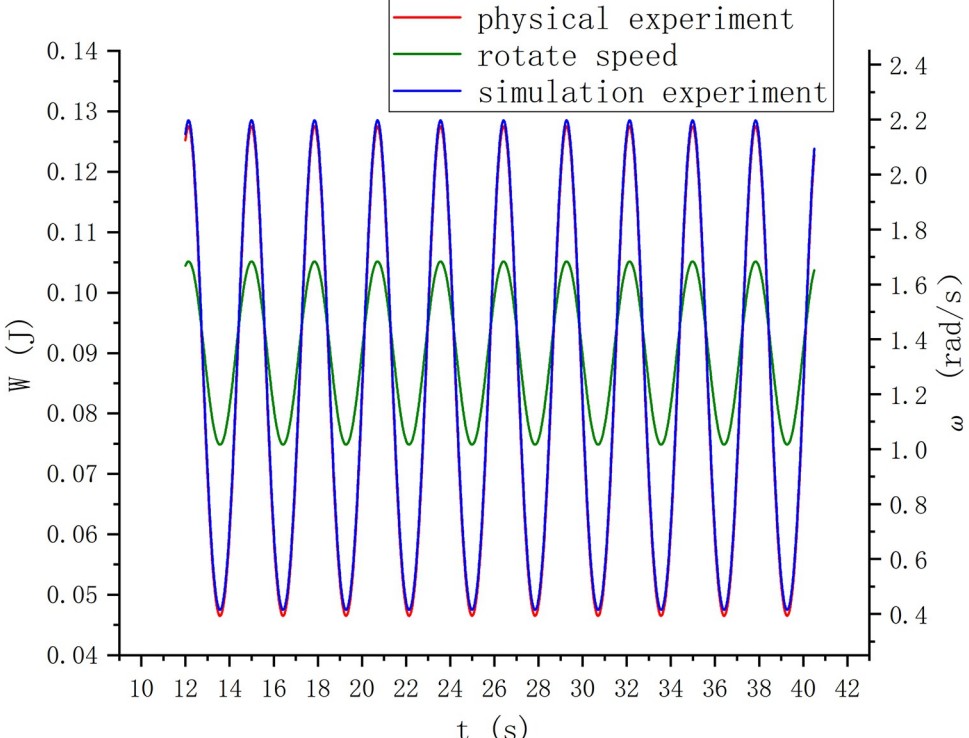

**Fig 18. Experimental verification diagram.**

type turbine and the study of application environment. The current research trend mainly starts from optimizing the structure and the applicable environment of the turbine. In this paper, we mainly discuss the installation optimization of S-type turbine, the operation condition in wave field environment, and the influence of wave field environmental parameters on S-type turbine. Based on the motion mechanism of S-type turbine in wave field and through numerical simulation and physical test, the following conclusions can be drawn.

1. It is found that the maximum energy captured by S-turbine with horizontal axis is 3 times that of vertical axis in wave field. On the wave side, the energy captured by the horizontal axis inverted S-turbine is more than 6 times that of the horizontal axis positive S-turbine. In wave energy capture, consideration of the installation of S-type turbines is a crucial factor. At the same time, as the wave height decreases, the captured wave energy also decreases.

**Table 5. Experimental parameter table.**

| Parameter | D (m) | h (m) | D/h |
|---|---|---|---|
| value | 0.2 | 0 | / |
| | | 0.03 | 6.67 |
| | | 0.06 | 3.33 |
| | | 0.09 | 2.22 |
| | | 0.12 | 1.67 |
| | 0.15 | 0.06 | 2.5 |
| | 0.2 | | 3.33 |
| | 0.25 | | 4.17 |
| | 0.3 | | 5 |

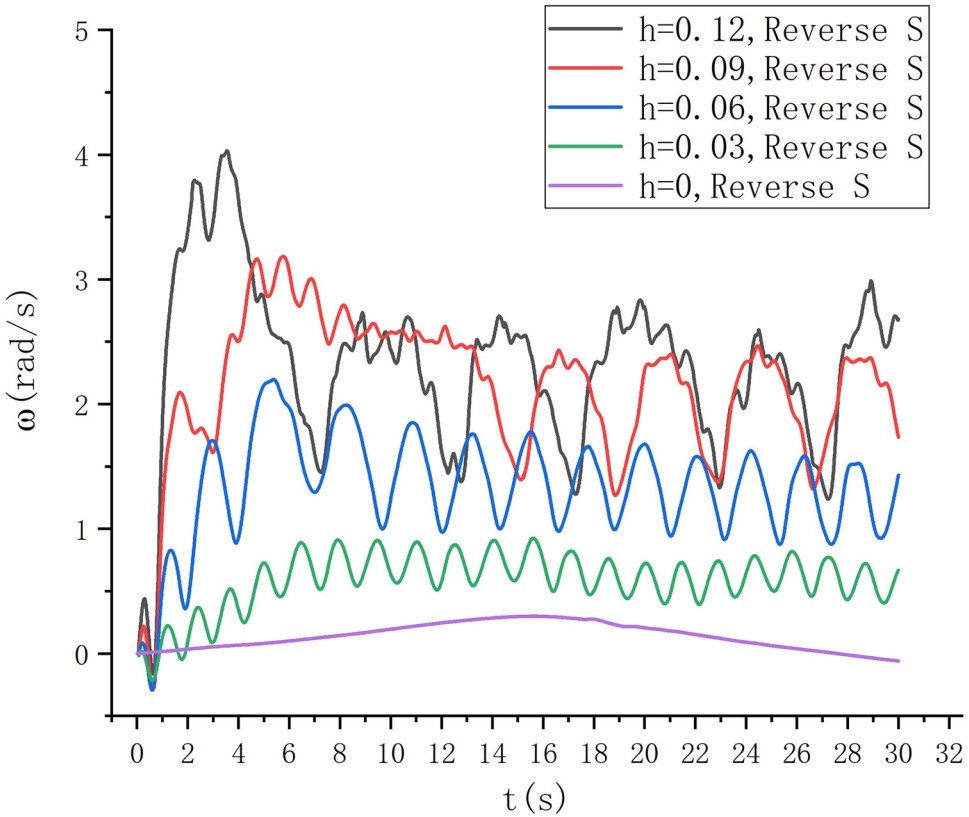

**Fig 19. Fixed water entry depth speed diagram.**

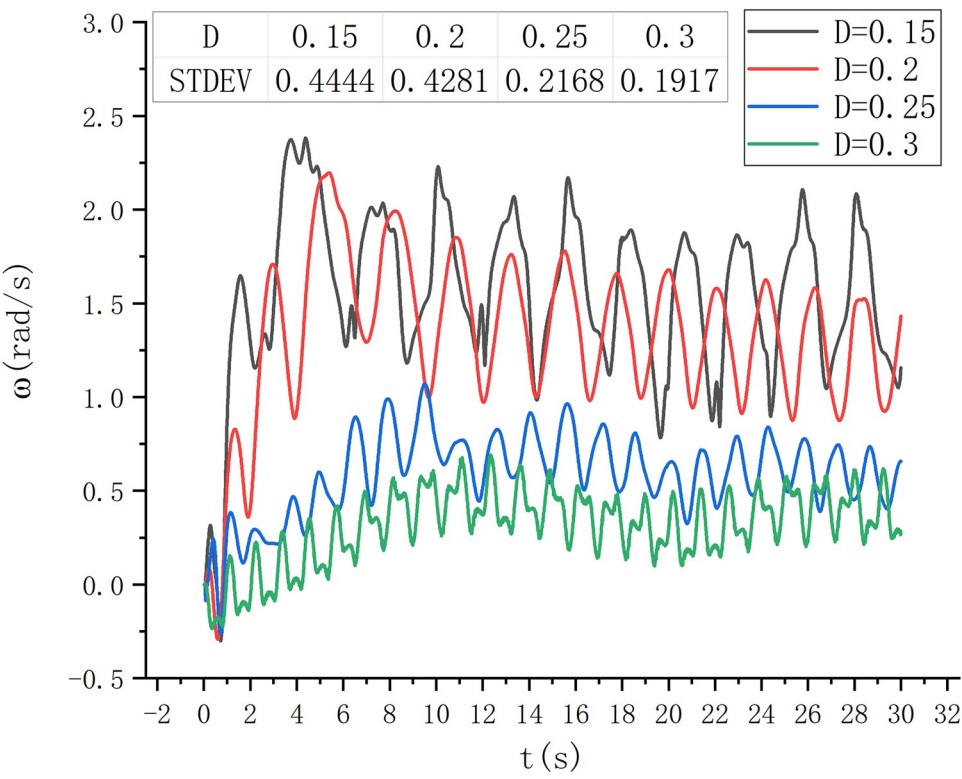

**Fig 20. Fixed wave height and rotational speed diagram.**

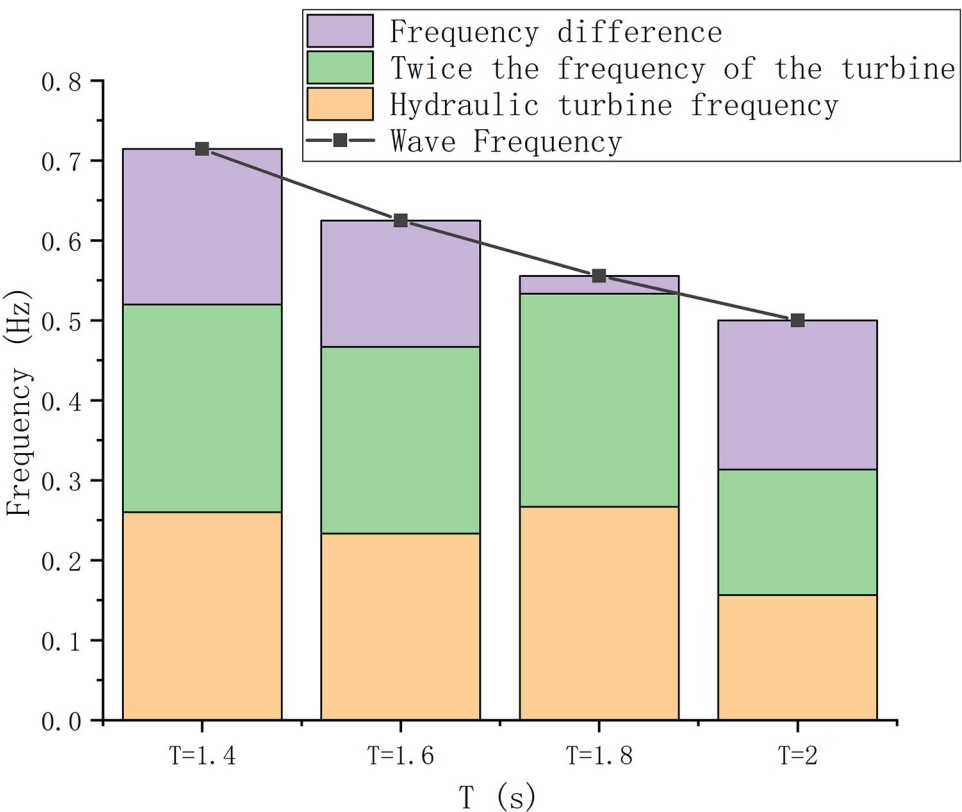

**Fig 21. Rotational frequency difference diagram of primary turbine.**

2. The speed of the S-turbine is related to the depth of entry relative to the wave height. The S-type turbine has a fluctuating velocity in the wave field. When the water depth is constant, the higher the wave height, the higher the speed of S-type turbine. As the wave height decreases, the S-turbine begins to rotate smoothly. When the wave height tends to 0, the speed of S-type turbine tends to 0, and the captured energy tends to 0. At the same time, because the S-turbine rotates at a pulsating speed, although a large blade tip speed ratio can be achieved instantaneously, the wave energy efficiency captured is not the maximum.

3. The higher the wave height, the more powerful the wave energy. The deeper you go, the stickier the water is. For the study of relative wave height and water entry depth D/h, both wave height and water entry depth will affect the rotation of S-type turbine. The S-type turbine rotates well when the wave height is 0.06 m and the water depth is 0.2 m. The dimensionless coefficient of the water depth relative to the wave height is about 3.33.

4. When the first-stage S-type turbine starts to rotate smoothly, the rotation of the S-type turbine is related to the wave period, the rotation frequency is close to 1/2 of the wave frequency. This rotation characteristic is related to its structure. The crest of each wave cycle impinges on the blades of the S-type turbine, causing the turbine to rotate.

Future studies need to consider other influencing factors. The actual ocean wave field is a kind of wave coupling field.It is necessary to consider the applicability of S-type turbine in a variety of wave fields. Water depth at high tide and low tide should be taken into account in the installation of S-turbine.

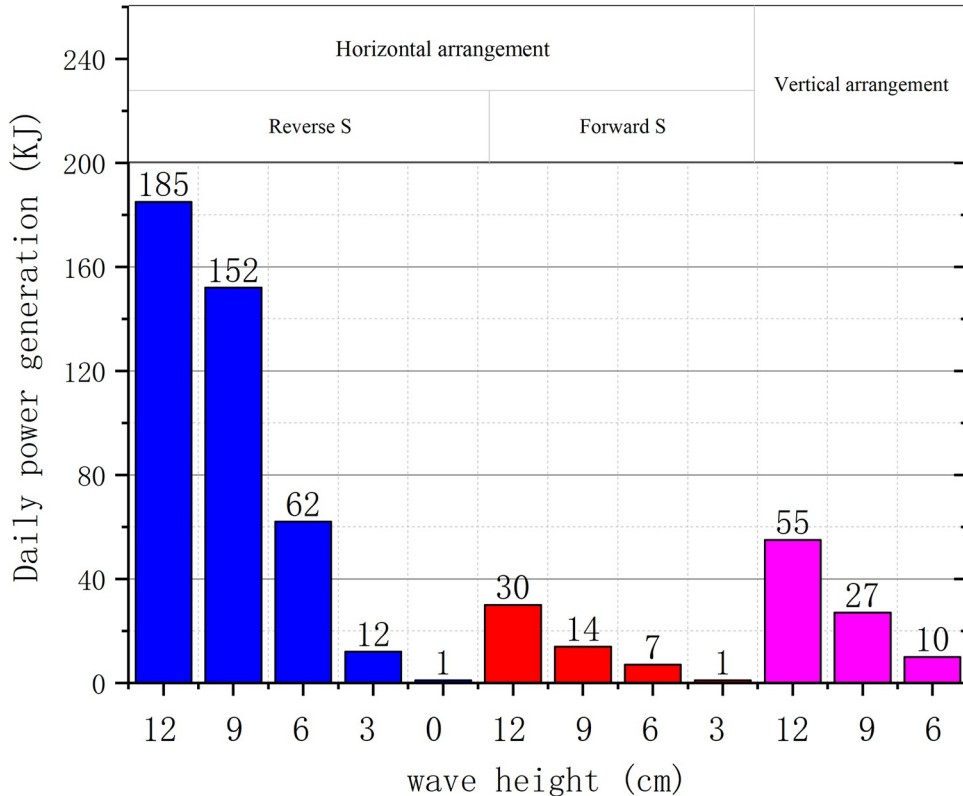

**Fig 22. Daily power generation.**

**Table 6. Performance table of S-type turbine.**

| Parameter | | h (m) | D/h | Max $C_p$ | Max TSR |
|---|---|---|---|---|---|
| Horizontal arrangement | Reverse S | 0 | / | 0.01 | 0.1 |
| | | 0.03 | 6.67 | 0.14 | 0.5 |
| | | 0.06 | 3.33 | 0.18 | 0.9 |
| | | 0.09 | 2.22 | 0.22 | 1 |
| | | 0.12 | 1.67 | 0.2 | 1 |
| | Forward S | 0.06 | 2.5 | 0.01 | 0.12 |
| | | | 3.33 | 0.02 | 0.32 |
| | | | 4.17 | 0.02 | 0.2 |
| | | | 5 | 0.03 | 0.15 |
| Vertical arrangement | | / | 0.06 | 3.33 | 0.1 | 0.6 |

## Supporting information

**S1 Data.**
(XLSX)

**S1 Nomenclature.**
(DOCX)

## Author Contributions

**Conceptualization:** ChuHua Jiang.

**Data curation:** Yue Zhuo.

**Formal analysis:** Yue Zhuo.

**Funding acquisition:** JunHua Chen.

**Investigation:** ChuHua Jiang.

**Methodology:** Ying Wang.

**Project administration:** JunHua Chen.

**Resources:** Hao Li.

**Software:** Hao Li.

**Validation:** Ying Wang, JunHua Chen.

**Visualization:** Yue Zhuo, Siru Wu.

**Writing – original draft:** LingJie Bao.

**Writing – review & editing:** LingJie Bao.

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
