## [Decision Letter · Decision Letter 0]

24 Jun 2024

PONE-D-24-12338Research on hydrodynamic performance of S-type Turbine based on linear wavePLOS ONE

Dear Dr. Wang,

Thank you for submitting your manuscript to PLOS ONE. After careful consideration, we feel that it has merit but does not fully meet PLOS ONE’s publication criteria as it currently stands. Therefore, we invite you to submit a revised version of the manuscript that addresses the points raised during the review process.A lot of literature is available on the subject. Author should discuss the finding of the earlier research and also discuss the limitations of those studies. The literature study can be enhanced by adding recent relevant references. Literature review should be improved and table should also be included which provide summary about the relevant previous research. Also, make the current literature review well structured.How you define the domain size? Will the wall disturb the flow?How you justify the value of y+ between 60 and 144? Is there any wall function?Rewrite section 3.2 in technical manners. Only mentioning the number of nodes/elements are not sufficient. Also explain the CFL value for this caseQuality of the results/graphs are not upto the mark. This should be improved.Novelty, choice of numerical model and meshing details are not explained well. Please submit your revised manuscript by Aug 08 2024 11:59PM. If you will need more time than this to complete your revisions, please reply to this message or contact the journal office at plosone@plos.org. Please include the following items when submitting your revised manuscript:A rebuttal letter that responds to each point raised by the academic editor and reviewer(s). You should upload this letter as a separate file labeled 'Response to Reviewers'.A marked-up copy of your manuscript that highlights changes made to the original version. You should upload this as a separate file labeled 'Revised Manuscript with Track Changes'.An unmarked version of your revised paper without tracked changes. You should upload this as a separate file labeled 'Manuscript'.

We look forward to receiving your revised manuscript.

Kind regards,

Niaz Bahadur Khan, PhD

Academic Editor

PLOS ONE

Journal Requirements:

3. Thank you for stating the following in your Competing Interests section: "NO authors have competing interests."

4. We note that your Data Availability Statement is currently as follows: "All relevant data are within the manuscript and its Supporting Information files."

Additional Editor Comments:

1. A lot of literature is available on the subject. Author should discuss the finding of the earlier research and also discuss the limitations of those studies. The literature study can be enhanced by adding recent relevant references. Literature review should be improved and table should also be included which provide summary about the relevant previous research. Also, make the current literature review well structured.

2. How you define the domain size? Will the wall disturb the flow?

3. How you justify the value of y+ between 60 and 144? Is there any wall function?

4. Rewrite section 3.2 in technical manners. Only mentioning the number of nodes/elements are not sufficient. Also explain the CFL value for this case

5. Quality of the results/graphs are not upto the mark. This should be improved.

6. Novelty, choice of numerical model and meshing details are not explained well.

Reviewers' comments:

Reviewer's Responses to Questions

**Comments to the Author**

1. Is the manuscript technically sound, and do the data support the conclusions?

Reviewer #1: Yes

Reviewer #2: Yes

2. Has the statistical analysis been performed appropriately and rigorously? 

Reviewer #1: Yes

Reviewer #2: Yes

3. Have the authors made all data underlying the findings in their manuscript fully available?

Reviewer #1: Yes

Reviewer #2: Yes

4. Is the manuscript presented in an intelligible fashion and written in standard English?

Reviewer #1: Yes

Reviewer #2: Yes

5. Review Comments to the Author

Reviewer #1: Presented paper investigated the hydrodynamic performance of a Savonius type turbine (S-type turbine) in a wave field.

The method of combining numerical simulation with physical experiment is adopted. Based on linear wave theory and turbulence model, Star CCM+ numerical simulation software is used for digital modelling, and overlapping grid technology is used for grid modelling.

The research results show that the S-type turbine must design a reasonable layout according to the wave field conditions.

Eq 4, how much is the moment of inertia of the turbine, I?

What is figure 5? it does not mean anything!

Table 3, what is TSR (Tip speed ratio)? or more detailed data of other related parameters? For example, U and inflow velocity or speed?

Fig 14 h is wave height, and wave period is T. In table 2 please show it.

Fig 17, obtain the average energy, how much for each case?

Fig 19, obtain the average values of the energy. omega is turbine angular velocity? Is that right?

Finally, compare the absorbed power for the forward and reverse turbine. Which one is better? What is the RPM? What is TSR? Add a table and show all average parameters of power, energy, etc corresponding to speed and TSR and some more else.

Reviewer #2: The work is presented on an interesting topic and fits the theme of PLOS ONE. There are several notes that authors need to take into account before publishing.

1. The background needs to be improved. In the analyzed works of other authors, the advantages and disadvantages of the reviewed studies should be presented. Based on the background results, it is necessary to clearly justify the choice of research tool. The goals and objectives of the study should be more clearly formulated.

2. Many questions on numerical modeling. There is no validation and verification process in your work. The choice of network is questionable, since the power factor is not a suitable value for determining the main parameters of turbine operation. The choice of the turbulence model is not justified in any way. No test results. There is no description of boundary conditions.

3. What is the error of the experimental data obtained?

How was it determined?

4. The design of the figures needs to be redesigned; the data curves should not fall on the numerical values of the axes.

5. Conclusions should be concise and clear, contain specific quantitative data, all discussions should be after the figures.

6. PLOS authors have the option to publish the peer review history of their article (what does this mean?). If published, this will include your full peer review and any attached files.

Reviewer #1: **Yes: **Hassan Ghassemi

Reviewer #2: No

---

## [Author Response · Author response to Decision Letter 0]

31 Jul 2024

1. A lot of literature is available on the subject. Author should discuss the finding of the earlier research and also discuss the limitations of those studies. The literature study can be enhanced by adding recent relevant references. Literature review should be improved and table should also be included which provide summary about the relevant previous research. Also, make the current literature review well structured.

Some changes have been made to the references. I made a table to show how innovative the research was. At the same time, it also shows the difference between S-type turbine under wave action and pure flow action. Mainly under the action of waves, the velocity size and direction of water particles are changed, while under the action of pure flow, it is considered that the velocity size and direction of water particles are unchanged. 

2. How you define the domain size? Will the wall disturb the flow?

According to the experimental rules of Chinese flume, a large enough distance is set before and after the S-type turbine to observe the experimental phenomenon of the S-type turbine. Symmetry planes are set on both sides of the fluid calculation domain, considering that the S-type turbine will be applied in the form of array in practical applications. Therefore, this setting will reduce the computation effort. Secondly, the gradient of physical quantities such as the normal velocity on the symmetric plane of the boundary conditions is 0. So it doesn't interfere with the flow of water. 

3. How you justify the value of y+ between 60 and 144? Is there any wall function?

The surface of S-type turbine needs wall treatment, and the fluid movement near the wall affected by it can be divided into three sub-layers: viscous bottom layer, logarithmic rate layer and buffer layer. The fluid in the viscous bottom layer is in contact with the wall surface, and the viscosity is dominant, almost laminar flow movement. The average flow rate is determined by fluid density, viscosity, wall distance and shear force. The fluid in the logarithmic rate layer is in the outermost layer, dominated by the flow shear stress, so that the turbulent state is fully developed, and the velocity distribution is close to the logarithmic rate. The effect of the viscous force on the transition layer between the buffer layer and the log-rate layer is similar to that of the turbulent shear stress, and the flow condition is complicated and the thickness is very small. 

When y+<5, it belongs to the viscous bottom layer, and the velocity is linearly distributed,u*=y. If 5<y+<30, it belongs to the buffer layer. When 30<y+<500, it belongs to the log-rate layer, and the velocity is distributed log-rate along the wall normal direction, u+=2.5In y+.

The value of y+ can be set as a function of y+ in a computational simulation, so the value of y+ can be observed between 60 and 140.

4. Rewrite section 3.2 in technical manners. Only mentioning the number of nodes/elements are not sufficient. Also explain the CFL value for this case

The simulation details are restated and the convergence residuals are less than 10−5

5. Quality of the results/graphs are not upto the mark. This should be improved.

The resulting graph has been updated.

6. Novelty, choice of numerical model and meshing details are not explained well.

The novelty of this paper is mainly the application of S-type turbine in wave field. Wave field is different from pure flow field. In a pure flow field, we think that the velocity magnitude and direction of the particle are constant at input, oriented, and change when encountering resistance or damping. In the wave field, the velocity and direction of water particles vary, as shown in Figure 3. In the mesh division, STAR-ccm+ software has a variety of mesh forms, polyhedral mesh, tetrahedral mesh, cut body mesh and other mesh forms. Compared with other grid forms, polyhedral mesh is more suitable for complex structure and mechanism motion. 

Reviewer #1: Presented paper investigated the hydrodynamic performance of a Savonius type turbine (S-type turbine) in a wave field.

The method of combining numerical simulation with physical experiment is adopted. Based on linear wave theory and turbulence model, Star CCM+ numerical simulation software is used for digital modelling, and overlapping grid technology is used for grid modelling.

The research results show that the S-type turbine must design a reasonable layout according to the wave field conditions.

1.Eq 4, how much is the moment of inertia of the turbine, I?

S-type turbine moment of inertia is (0.05, 0.09,0.11)

2.What is figure 5? it does not mean anything!

Figure 5 has been deleted.

3.Table 3, what is TSR (Tip speed ratio)? or more detailed data of other related parameters? For example, U and inflow velocity or speed?

Tip velocity ratio is also an important index to measure turbine performance. The author originally considered that in the wave field, the speed and direction of the motion of water particles are not fixed, and wanted to use the power coefficient to characterize the performance of the turbine. At the suggestion of the reviewer, the maximum blade tip ratio can be used to characterize the performance index of the turbine. 

4.Fig 14 h is wave height, and wave period is T. In table 2 please show it.

Symbol indicated

5.Fig 17, obtain the average energy, how much for each case?

The author further explains in Figure 23, and Figure 17 shows the energy of the kinetic energy of the turbine under the action of the wave, because its speed is not stable under the action of the wave. This is also what the author wants to reveal the details of the movement of the S-type turbine under the action of waves. 

6.Fig 19, obtain the average values of the energy. omega is turbine angular velocity? Is that right?

The energy obtained is the steady rotation of the S-turbine, but this energy is fluctuating. Specifically, the kinetic energy of the S-turbine, if the energy is to be used, it needs to be converted into electricity. This paper mainly studies the first-order S-type turbine. If it is replaced with second-order S-type turbine or third-order S-type turbine, the energy curve will be smoother. This is what our team will work on next. 

7.Finally, compare the absorbed power for the forward and reverse turbine. Which one is better? What is the RPM? What is TSR? Add a table and show all average parameters of power, energy, etc corresponding to speed and TSR and some more else.

The authors added tables to the results discussion.

Reviewer #2:The work is presented on an interesting topic and fits the theme of PLOS ONE. There are several notes that authors need to take into account before publishing.

1. The background needs to be improved. In the analyzed works of other authors, the advantages and disadvantages of the reviewed studies should be presented. Based on the background results, it is necessary to clearly justify the choice of research tool. The goals and objectives of the study should be more clearly formulated.

In this paper, the S-type turbine is further discussed, and the table is added to facilitate the author to read and understand. 

2. Many questions on numerical modeling. There is no validation and verification process in your work. The choice of network is questionable, since the power factor is not a suitable value for determining the main parameters of turbine operation. The choice of the turbulence model is not justified in any way. No test results. There is no description of boundary conditions.

Perhaps our description of the work is not enough, STAR-ccm+ software provides a relatively rich grid forms, such as polyhedral mesh, tetrahedral mesh, cut body mesh, etc. The S-type turbine we studied adopts polyhedral mesh, mainly for the complex structure and movement. The Renomean Nevistox k-e model is selected because it meets most of the calculation requirements, and there is no strong swirl, buoyant flow, neutral stratified flow and other conditions in the S-type turbine. k-e,RNG k-e and Realizable k-e models have been used by many researchers to study the rotation of S-type turbines. 

3. What is the error of the experimental data obtained?

How was it determined?

The S-type turbine is an unstable rotation under the action of waves, and there are data fluctuations and randomness, which will result in some experimental errors. Therefore, the author uses the method of averaging to calculate the energy capture power of the turbine, and the results are further discussed in the fifth part. 

4. The design of the figures needs to be redesigned; the data curves should not fall on the numerical values of the axes.

The diagram has been redrawn.

5. Conclusions should be concise and clear, contain specific quantitative data, all discussions should be after the figures.

The author has recalibrated the conclusions.

---

## [Decision Letter · Decision Letter 1]

2 Sep 2024

Research on hydrodynamic performance of S-type Turbine based on linear wave

PONE-D-24-12338R1

Dear Dr. Wang,

We’re pleased to inform you that your manuscript has been judged scientifically suitable for publication and will be formally accepted for publication once it meets all outstanding technical requirements.

Kind regards,

Niaz Bahadur Khan, PhD

Academic Editor

PLOS ONE

Additional Editor Comments (optional):

Reviewers' comments:

Reviewer's Responses to Questions

**Comments to the Author**

1. If the authors have adequately addressed your comments raised in a previous round of review and you feel that this manuscript is now acceptable for publication, you may indicate that here to bypass the “Comments to the Author” section, enter your conflict of interest statement in the “Confidential to Editor” section, and submit your "Accept" recommendation.

Reviewer #1: All comments have been addressed

Reviewer #2: All comments have been addressed

2. Is the manuscript technically sound, and do the data support the conclusions?

Reviewer #1: Partly

Reviewer #2: Yes

3. Has the statistical analysis been performed appropriately and rigorously? 

Reviewer #1: Yes

Reviewer #2: Yes

4. Have the authors made all data underlying the findings in their manuscript fully available?

Reviewer #1: Yes

Reviewer #2: Yes

5. Is the manuscript presented in an intelligible fashion and written in standard English?

Reviewer #1: Yes

Reviewer #2: Yes

6. Review Comments to the Author

Reviewer #1: This is good topic and good paper. It is revised well and can be accepted. In some cases, there are some questionable but it is not important.

Reviewer #2: The explanations and changes made by the authors to the work dispelled some of my misconceptions. The article looks like an organic finished product for publication.

7. PLOS authors have the option to publish the peer review history of their article (what does this mean?). If published, this will include your full peer review and any attached files.

Reviewer #1: **Yes: **Hassan Ghassemi

Reviewer #2: No

---

## [Editor Report · Acceptance letter]

31 Oct 2024

PONE-D-24-12338R1 

PLOS ONE

Dear Dr. Wang, 

I'm pleased to inform you that your manuscript has been deemed suitable for publication in PLOS ONE. Congratulations! Your manuscript is now being handed over to our production team.

Kind regards, 

on behalf of

Dr Niaz Bahadur Khan 

Academic Editor

PLOS ONE